# Vitamin D, B9, and B12 Deficiencies as Key Drivers of Clinical Severity and Metabolic Comorbidities in Major Psychiatric Disorders

**DOI:** 10.3390/nu17071167

**Published:** 2025-03-27

**Authors:** Mélanie Faugere, Éloïse Maakaron, Vincent Achour, Pierre Verney, Christelle Andrieu-Haller, Jade Obadia, Guillaume Fond, Christophe Lançon, Théo Korchia

**Affiliations:** 1Department of Academic Psychiatry, Sainte Marguerite University Hospital, Assistance Publique des Hôpitaux de Marseille, 13009 Marseille, France; eloise.maakaron@ap-hm.fr (É.M.); vincent.achour@ap-hm.fr (V.A.); pierre.verney@ap-hm.fr (P.V.); christelle.andrieu@ap-hm.fr (C.A.-H.); jade.obadia@ap-hm.fr (J.O.); guillaume.fond@gmail.com (G.F.); christophe.lancon@ap-hm.fr (C.L.); theo.korchia@ap-hm.fr (T.K.); 2Assistance Publique des Hôpitaux de Marseille, Aix-Marseille University, UR3279: Health Service Research and Quality of Life Center—CEReSS, 13005 Marseille, France; 3Groupement de Coopération Sanitaire, Centre de Recherche en Santé Mentale et Psychiatrie de la Région PACA, 13100 Aix en Provence, France; 4Service du Pr Christophe Lançon, CHU Sainte Marguerite, Pavillon Solaris, 270 Boulevard de Sainte Marguerite, 13009 Marseille, France

**Keywords:** vitamin D deficiency, folate, vitamin B12, schizophrenia, depression, bipolar disorder, psychiatric symptoms, metabolic syndrome, nutritional psychiatry

## Abstract

**Background/Objectives**: Severe mental illnesses such as schizophrenia, major depressive disorder, and bipolar disorder are often accompanied by metabolic comorbidities. While the role of vitamins in physical health is well-established, their involvement in psychiatric disorders has garnered increasing attention in recent years. **Methods**: We conducted a cross-sectional analysis of 1003 patients diagnosed with severe mental illnesses. Vitamin D, B9, and B12 serum levels were measured, and deficiencies were defined using established clinical cutoffs. Multivariate regression analyses were performed to identify associations between vitamin deficiencies and clinical outcomes. **Results**: Our findings revealed that vitamin deficiencies were prevalent across all diagnostic groups, with particularly high rates in patients with schizophrenia and major depressive disorder. Vitamin D deficiency was significantly associated with worse psychiatric outcomes, including increased depressive symptoms (adjusted OR = 1.89, *p* = 0.018), lower Global Assessment of Functioning scores (adjusted OR = −0.18, *p* < 0.001), and higher rates of metabolic syndrome (adjusted OR = 1.97, *p* = 0.007). Folate and B12 deficiencies were also linked to greater psychiatric symptom severity and metabolic disturbances, including increased risks of obesity and dyslipidemia. **Conclusions**: Our study highlights the critical role of vitamins deficiencies in both psychiatric and metabolic health of patients with severe mental illnesses. These findings underscore the importance of routine screening and correction of these deficiencies as part of comprehensive care in psychiatric populations. The integration of nutritional interventions may offer a novel and holistic approach to improving both mental and physical health outcomes.

## 1. Introduction

Major psychiatric disorders such as schizophrenia (SZ), major depressive disorder (MDD), and bipolar disorder (BD) are debilitating conditions that affect a significant proportion of the global population [1]. These disorders are characterized by profound alterations in cognitive [2], emotional [3], and behavioral functions [4], leading to a significant reduction in patients’ quality of life [5] and imposing a substantial economic burden on healthcare systems [6,7]. While the management of these conditions relies primarily on pharmacological [8] and psychotherapeutic [9] interventions, increasing attention has been paid to the role of nutritional factors in the etiology and progression of psychiatric illnesses [10].

Among these factors, vitamins D, B9 (folate), and B12 have attracted particular interest due to their involvement in neurological function [11] and potential association with mental health [12]. Vitamin D, traditionally known for its role in calcium metabolism and bone health [13], has also been identified as a key regulator of brain function [14,15], influencing processes such as neuroinflammation [16], synaptic plasticity [17], and neurogenesis [18]. Several epidemiological studies have suggested that low vitamin D levels may be associated with an increased risk of developing psychiatric disorders [19,20,21], including depression, schizophrenia, and bipolar disorder. In addition, there is recent evidence that vitamin D has a protective effect on major depressive disorder [22]. Schizophrenia and bipolar disorder were significantly associated with lower vitamin D concentrations [23]. However, in schizophrenia, there have been very few studies investigating vitamin D supplementation, and additionally, they have all been based on small samples [24].

Vitamins B9 (folate) and B12 are essential for the metabolism of homocysteine, an amino acid whose excessive accumulation has been linked to neurotoxicity [25], an increased risk of cognitive dysfunction [26], and psychiatric conditions [27]. Folate and vitamin B12 deficiencies have been frequently observed in patients with depression [28,29] and schizophrenia [30]. On the other hand, elevated serum vitamin B12 levels have been suggested to increase the risk of bipolar disorder [31]. These findings suggest a possible role for these vitamins in the pathophysiology of these disorders. However, the underlying mechanisms and the extent of their impact on mental health remain incompletely understood.

The growing interest in these vitamins within the field of psychiatry is also driven by their therapeutic potential. Several studies have investigated the efficacy of vitamin D or B vitamin supplementation in psychiatric patients [32], in MDD [33,34,35], and in schizophrenia [36,37], with promising preliminary results. However, current data remain heterogeneous, and few studies have systematically evaluated the combined and specific effects of these vitamins on different clinical parameters in psychiatric disorders.

Furthermore, vitamin D, B9, and B12 deficiencies may also be associated with cardiovascular and metabolic risk factors, such as obesity [38], metabolic syndrome [39], hypertension [40], and dyslipidemia [41], which are often observed in patients with schizophrenia [42], depression [43], or bipolar disorder [44]. These comorbidities, which often worsen the prognosis of psychiatric disorders, may themselves be influenced by vitamin deficiencies, highlighting the importance of a holistic approach that integrates both psychiatric and metabolic dimensions in the study of these populations.

### Rationale and Study Objective

In this context, it is essential to better understand how levels of vitamins D, B9, and B12 influence not only the onset and progression of psychiatric disorders, but also their relationship with specific clinical variables, such as symptom severity, quality of life, and metabolic comorbidities. Although several studies have identified significant deficiencies in these vitamins in patients with psychiatric disorders [45], few have comprehensively analyzed their combined and specific effects on multidimensional clinical parameters.

The present study aims to fill this gap by investigating the role of vitamins D, B9, and B12 in a range of clinical and metabolic variables in patients diagnosed with SZ, MDD, or BD. Using a monocentric dataset, we sought to identify correlations between vitamin levels and parameters such as symptom severity, functional status, quality of life, and the presence of metabolic comorbidities. We also assessed the potential impact of these vitamin deficiencies on the clinical course and prognosis of patients. This study contributes to a better understanding of the complex interactions between nutrition, metabolism, and mental health and may provide new avenues for therapeutic interventions.

## 2. Materials and Methods

### 2.1. Study Design and Population

This was a cross-sectional observational study conducted in a specialized psychiatric center in Marseille, France. The study included adult patients diagnosed with schizophrenia (SZ), major depressive disorder (MDD), or bipolar disorder (BD) according to the criteria of the Diagnostic and Statistical Manual of Mental Disorders, 5th Edition (DSM-5) [46]. All the participants were recruited between 2018 and 2023, during either specialized psychiatric consultations or hospitalization at the center.

The data used in this study are part of a monocentric database, systematically constructed to evaluate patients treated for these psychiatric disorders. This database contains demographic, clinical, and biological information that allows the study of the relationships between vitamin levels (D, B9, and B12) and various clinical and metabolic parameters.

### 2.2. Inclusion and Exclusion Criteria

Inclusion criteria:-Age between 18 and 65 years;-Primary diagnosis of SZ, MDD, or BD;-Ongoing treatment or follow-up at the psychiatric center for these disorders;-Signed informed consent for participation and use of clinical data for research purposes.

Exclusion criteria:-Current pregnancy or intention to become pregnant within 6 months of inclusion;-Serious, unstable chronic diseases or neurological conditions that could alter the results (e.g., autoimmune diseases or cancer);-Regular use of vitamin D, B9, or B12 supplements within three months prior to inclusion;-Inability to complete clinical assessments due to severe cognitive impairment or acute psychiatric illness.

### 2.3. Data Collection and Clinical Scales

Clinical data were collected during structured medical consultations and from medical records and standardized interviews. Sociodemographic data such as age, sex, employment status, and education level were recorded, as well as psychiatric and somatic history. Clinical scales were administered by trained health professionals. Each patient completed the questionnaires under the supervision of a clinician to ensure understanding and accuracy of responses. Data quality was ensured through random data checks, training of personnel, and supervised completion of questionnaires to ensure accurate understanding of the questions.

Blood samples were taken to measure serum levels of vitamins D, B9, and B12. Vitamins were quantified in the hospital laboratory using a commercial radioimmunoassay (electrochemiluminescence method, COBAS 6000). Vitamin D deficiency was defined as a serum level below 25 ng/mL, vitamin B9 deficiency as a level below 10.4 nmol/L, and vitamin B12 deficiency as a level below 140 pg/mL. All biological analyses were coordinated and performed by the same laboratory to ensure consistency and accuracy of the results. Threshold values were defined using the local laboratory standards according to the World Health Organization definitions.

### 2.4. Clinical Scales

The patients were assessed using several validated clinical scales to quantify the severity of psychiatric symptoms and functional outcomes:-Depressive symptoms: the Calgary Depression Scale for Schizophrenia (CDSS) [47], recently validated for mood disorders [48], was used to assess depressive symptoms in SZ patients, with a score ≥6 indicating clinical depression.-Anxiety symptoms: anxiety was measured using the Spielberger State-Trait Anxiety Inventory (STAI-YA) [49], with a score ≥40 indicating clinically significant anxiety [50].-Global functioning: the Global Assessment of Functioning (GAF) scale [51] was used to measure overall functioning, with a score <50 indicating severe functional impairment.-Quality of life: the Short Form-36 (SF-36) [52] was used to assess quality of life in two dimensions, physical and mental. A score below the general population average [50] was considered to indicate impaired quality of life.-Suicide risk: the Suicidal Behavior Questionnaire-Revised (SBQ-R) [53] was used to assess suicide risk. A score ≥8 was interpreted as indicating a high risk of suicide.-Metabolic syndrome: the International Diabetes Federation (IDF) criteria [54] were used to diagnose metabolic syndrome, based on factors such as high waist circumference, hypertension, hyperglycemia, elevated triglycerides, and low HDL cholesterol levels.-Metabolic comorbidities, such as obesity, hypertension, and dyslipidemia, were recorded based on biological data and medical records. The patients were also assessed for substance use, including tobacco (using the Fagerström Test for Nicotine Dependence [55]), alcohol, and cannabis.

### 2.5. Statistical Analysis

Statistical analyses were performed using SPSS software (version 20.0). All variables were presented using means and dispersion (standard deviation) for continuous variables and frequency distribution for categorical variables. Comparisons between individuals with and without hypovitaminosis (D, B9, or B12) were made using the chi-squared test for categorical variables. Continuous variables were analyzed with Student’s *t*-test.

Multiple regression analyses were performed to identify factors associated with vitamin deficiencies and clinical or metabolic variables, adjusted for age and sex, and with statistical significance set at *p* < 0.05.

## 3. Results

A total of 1003 patients were included in the study, divided into three diagnostic groups: schizophrenia (SZ) (n = 463), major depressive disorder (MDD) (n = 427), and bipolar disorder (BD) (n = 113). We assessed the prevalence of deficiencies in vitamins D, B9 (folate), and B12 across these groups, along with their associations with various clinical, functional, and metabolic parameters. Only results that remained significant in multivariate analyses, after adjusting for age and sex, are presented.

### 3.1. Vitamin D Deficiency Across Diagnostic Groups

Vitamin D deficiency, or hypovitaminosis D, was prevalent in all three psychiatric populations, though its clinical impact varied across diagnoses. In schizophrenia (SZ) (Table 1), 19.9% of patients (n = 92) exhibited vitamin D deficiency. This deficiency was strongly associated with poorer clinical and functional outcomes, including lower employment rates (adjusted odds ratio (aOR) = 0.205, *p* = 0.009), increased agoraphobia (aOR = 3.417, *p* < 0.001), and higher depressive symptoms (aOR = 1.894, *p* = 0.018). Functional impairments, as measured by the Global Assessment of Functioning (GAF) score, were significantly worse in patients with vitamin D deficiency (aOR = −0.184, *p* < 0.001), while higher clinical global impression (CGI) scores (aOR = 0.098, *p* = 0.043) were indicative of greater illness severity. Metabolic complications were also prominent, with hypovitaminosis D linked to obesity (aOR = 1.789, *p* = 0.028), hypertriglyceridemia (aOR = 2.171, *p* = 0.002), and metabolic syndrome (aOR = 1.969, *p* = 0.009).

Among the major depressive disorder (MDD) patients (Table 2), 11.7% (n = 50) were found to have vitamin D deficiency. Hypovitaminosis D in this group was associated with decreased physical health, as measured by the SF-36 physical health score (aOR = −0.150, *p* = 0.012), as well as higher levels of systemic inflammation (hsCRP, aOR = 0.148, *p* = 0.003). Patients with vitamin D deficiency also exhibited a higher likelihood of agoraphobia (aOR = 2.930, *p* = 0.001) and hypertriglyceridemia (aOR = 2.251, *p* = 0.020).

In bipolar disorder (BD) (Table 3), 8.8% of the patients (n = 10) exhibited vitamin D deficiency. This deficiency was significantly associated with older age (aOR = 0.202, *p* = 0.032) and the use of atypical antipsychotics (aOR = 4.512, *p* = 0.035), suggesting a potential interaction between antipsychotic medications and vitamin D metabolism. The association with metabolic issues was less pronounced in this population, but it remains a point of concern for future studies.

### 3.2. Vitamin B9 (Folate) Deficiency

Folate deficiency was a common finding in our cohort, affecting over a quarter of patients in both SZ and MDD groups, with a slightly lower prevalence in BD.

In schizophrenia (SZ) (Table 4), 27.2% (n = 82) of patients were deficient in vitamin B9. Multivariate analysis revealed that folate deficiency was more common in male patients (aOR = 0.498, *p* = 0.035) and those with lower educational attainment (aOR = 0.401, *p* = 0.005). Interestingly, higher folate levels were associated with a reduced likelihood of panic disorder (aOR = 0.386, *p* = 0.040) and social phobia (aOR = 0.364, *p* = 0.029), suggesting a potential protective effect of folate in anxiety disorders.

Among the MDD patients (Table 5), 27.9% (n = 88) exhibited vitamin B9 deficiency. Younger age was a significant predictor of folate deficiency (aOR = −0.245, *p* < 0.001), along with the presence of ADHD (aOR = 2.082, *p* = 0.034). The strong association between lower folate and vitamin B12 levels (aOR = −0.150, *p* = 0.011) highlights the interdependency of these nutrients, which is particularly important in understanding their role in psychiatric disorders.

In bipolar disorder (BD) (Table 6), 22.6% (n = 19) of the patients had folate deficiency. While multivariate analysis did not yield significant clinical associations in this group, the prevalence remains notable and warrants further investigation.

### 3.3. Vitamin B12 Deficiency

Vitamin B12 deficiency was relatively rare across all the diagnostic groups, with 1.0% (n = 3) in SZ, 3.2% (n = 10) in MDD, and 2.4% (n = 2) in BD. Despite the low prevalence, B12 deficiency was significantly associated with older age in both SZ (aOR = 0.116, *p* = 0.045) and BD (aOR = 0.202, *p* = 0.032). Furthermore, in the SZ group, vitamin B12 deficiency was linked to agoraphobia (aOR = 12.893, *p* = 0.041), albeit the small sample size limits the generalizability of these findings (Appendix A).

Given the low prevalence of B12 deficiency, its direct impact on clinical outcomes in this cohort was limited. However, its interplay with folate and vitamin D levels should not be overlooked, as combined deficiencies may exacerbate psychiatric and metabolic symptoms.

## 4. Discussion

Our study highlights the crucial and multifaceted role that in vitamin D, B9 (folate), and B12 deficiencies play in the psychiatric and metabolic health of patients with schizophrenia (SZ), major depressive disorder (MDD), and bipolar disorder (BD). While these micronutrients are often undervalued in psychiatric care, their profound impact on neuroinflammation [16], neurotransmission [17], and overall metabolic homeostasis [38] underscores their importance as modifiable therapeutic targets. By addressing these deficiencies, we can introduce a new paradigm in the treatment of severe mental illnesses—one that integrates nutritional interventions alongside standard psychopharmacological and psychosocial therapies.

### 4.1. Vitamin D: A Neurosteroid with Systemic Influence

The role of vitamin D in mental health has shifted from that of a simple calcium-regulating agent to that of a neurosteroid with extensive involvement in brain function [56]. Beyond its traditional role in bone homeostasis [14], vitamin D has now been implicated in the modulation of key neurological pathways, including neuroinflammation [16], neurotransmitter regulation [17], neuroplasticity [18] and DNA methylation [57,58]. These processes are fundamental to maintaining cognitive and emotional balance [59,60], which are often impaired in psychiatric disorders such as SZ, MDD, and BD.

Neuroinflammation has emerged as a core feature of many psychiatric disorders, particularly schizophrenia [61] and depression [62], where pro-inflammatory cytokines such as IL-6, TNF-alpha, and C-reactive protein are persistently elevated [63]. Chronic inflammation disrupts synaptic plasticity, neurotransmitter release, and neural circuit function, contributing to cognitive deficits [64] and emotional dysregulation [65] observed in these disorders. The anti-inflammatory properties of vitamin D, in particular, its ability to suppress pro-inflammatory cytokines while promoting anti-inflammatory mediators such as IL-10, make it a potent modulator of the neuroimmune response [66]. By attenuating this inflammatory cascade, vitamin D may help to preserve synaptic integrity and enhance neuroplasticity, thereby alleviating some of the cognitive and emotional impairments associated with psychiatric disorders.

In addition, the role of vitamin D in regulating neurotransmitter systems, particularly glutamatergic and GABAergic signaling, is critical [67]. Disruptions in these pathways are central to the pathophysiology of schizophrenia, where an imbalance between excitatory and inhibitory neurotransmission contributes to cognitive impairments and negative symptoms [68]. Vtitamin D’s ability to modulate glutamate toxicity and enhance GABAergic signaling may explain its beneficial effects on cognitive performance in schizophrenia and other psychotic disorders [69]. In addition, vitamin D is involved in the synthesis of serotonin [70], which plays a key role in mood regulation. The serotonergic effects of vitamin D supplementation could, therefore, alleviate depressive symptoms, particularly in MDD [71].

Furthermore, vitamin D status has a significant influence on DNA methylation [72], and researchers have demonstrated that gestational vitamin D deficiency modulates DNA methylation of depression-related genes in mice [73], that DNA methylation could predict antenatal and postpartum depression among women [74], and that DNA methylation has been demonstrated as a biomarker in patients with hypovitaminosis D and anxiety or depression [75].

The systemic influence of vitamin D extends to metabolic regulation, an important consideration for psychiatric populations. Patients with SZ and BD are at an increased risk of developing metabolic syndrome [76], a cluster of conditions including obesity, insulin resistance, dyslipidemia, and hypertension. These metabolic abnormalities are often exacerbated by the use of antipsychotic medications [77], which disrupt glucose and lipid metabolism. The role of vitamin D in improving insulin sensitivity and regulating lipid profiles offers a potential intervention to mitigate the metabolic side effects of psychiatric medications [78]. By correcting vitamin D deficiency, clinicians may be able to reduce the incidence of metabolic syndrome and ultimately lower the risk of cardiovascular diseases and diabetes in these vulnerable populations.

### 4.2. Folate (B9) and Vitamin B12: Gatekeepers of Neurotransmission and Cognitive Health

Folate and vitamin B12 are essential cofactors in the one-carbon metabolic cycle, which is responsible for DNA methylation, neurotransmitter synthesis, and the regulation of homocysteine levels [79]. This cycle is essential for maintaining proper brain function, and its disruption by vitamin deficiency can lead to significant cognitive and emotional impairments [80]. In psychiatric populations, elevated homocysteine levels resulting from B9 and B12 deficiency have been associated with neurotoxicity [81], oxidative stress [82], and vascular dysfunction [83]—all of which contribute to cognitive decline and mood dysregulation observed in SZ and MDD.

Folate plays a particularly important role in the synthesis of serotonin, dopamine, and norepinephrine [84]—neurotransmitters that are crucial for mood regulation. Low folate levels disrupt the production of these neurotransmitters, leading to the characteristic symptoms of depression, including anhedonia, low energy, and emotional instability. The relationship between folate deficiency and depression is well-established [85,86], with studies consistently showing that patients with low folate levels are more likely to experience treatment-resistant depression [87]. This suggests that folate supplementation could improve the efficacy of antidepressant medications, particularly in people who do not respond to conventional treatments.

Vitamin B12, on the other hand, is essential for maintaining neuronal integrity through its role in myelin synthesis and repair [88]. Myelin is the protective sheath that surrounds neurons and facilitates efficient signal transmission. In B12 deficiency, demyelination occurs, leading to slowed cognitive processing, memory impairment, and executive dysfunction [79]. These cognitive deficits are particularly prominent in schizophrenia, where they contribute to poor functional outcomes and quality of life. Restoring B12 levels may improve cognitive performance and reduce the severity of negative symptoms in schizophrenia [30], offering a promising adjunctive treatment strategy.

Additionally, both folate and B12 deficiencies increase homocysteine levels [38], which are strongly associated with neuroinflammation and vascular dysfunction. Elevated homocysteine is a well-documented risk factor for cognitive decline, particularly in aging populations [89], and has been implicated in the development of neurodegenerative diseases such as Alzheimer’s disease [90]. In psychiatric populations, elevated homocysteine levels can exacerbate cognitive impairment [91] and mood disorders [92]. Reducing homocysteine levels through targeted vitamin supplementation may mitigate some of the neurotoxic effects associated with psychiatric disorders, potentially improving both cognitive and emotional outcomes [93].

### 4.3. Nutrients and Clinical Outcomes

Hypovitaminosis is known to be associated with several clinical manifestations that appear to be ameliorated [94,95] or even prevented [33,35,96] by vitamin supplementation.

Low vitamin concentrations have been frequently associated with depression and anxiety symptoms [97,98], neuropsychiatric manifestations (such as cognitive or sleep symptoms) [99], but also with psychotic symptoms [100,101] and cognition [102,103].

These deficits have also been found in several other conditions, such as schizophrenia [104], tobacco use disorder [105], or attention deficit and hyperactivity disorder [106].

### 4.4. Integrating Nutritional Interventions into Psychiatric Care

The therapeutic potential of addressing these vitamin deficiencies is substantial, particularly when integrated into standard psychiatric care. Correcting vitamin D, B9, and B12 deficiencies offers a dual-target strategy: improving psychiatric symptoms while simultaneously addressing the physical comorbidities that are common in these populations [107]. Nutritional interventions should therefore be seen not as an adjunct but rather as an integral part of comprehensive psychiatric care.

In depression, several clinical trials have shown that vitamin D and folate supplementation can improve the response to antidepressants [108], particularly in people with treatment-resistant depression [109]. These interventions are thought to work by modulating serotonergic pathways and reducing the inflammatory markers that are often elevated in depressed patients [110]. In schizophrenia, where negative symptoms and cognitive deficits are notoriously resistant to antipsychotic treatment [111], nutritional interventions targeting B9 and B12 deficiencies offer a promising alternative [112]. By improving neurotransmitter synthesis [17,84], enhancing neuroplasticity [18], and reducing neuroinflammation [16,66], these vitamins may improve functional outcomes and quality of life for patients with schizophrenia.

Equally important is the impact of these nutritional interventions on metabolic health. Psychiatric patients, particularly those taking antipsychotic medication, are at high risk of developing metabolic syndrome [76], which significantly increases their risk of cardiovascular disease and diabetes. By addressing vitamin D deficiency, clinicians can reduce the metabolic complications associated with antipsychotic treatment, thereby improving long-term health outcomes. Integrating nutritional psychiatry into standard care could, therefore, have far-reaching benefits, not only for mental health, but also for reducing the overall burden of disease in psychiatric populations.

### 4.5. Strengths and Limitations

A key strength of our study is its comprehensive examination of the dual role of vitamins D, B9, and B12 in both psychiatric and metabolic health. By focusing on patients with severe mental illness, our study provides clinically relevant findings that can be directly translated into therapeutic interventions. The use of standardized biochemical assays to measure serum vitamin levels enhances the reliability of our findings, reducing measurement bias and ensuring robust conclusions about the association between vitamin deficiencies and psychiatric outcomes.

In addition, the integration of psychiatric and metabolic health outcomes positions our study at the intersection of two critical areas of health care, providing a holistic view of patient health. This dual focus allows for the exploration of integrated care models that could simultaneously address mental and physical health, leading to improved long-term outcomes for psychiatric patients.

However, the cross-sectional design of our study limits the ability to establish causality between vitamin deficiencies and psychiatric or metabolic outcomes. Although we found significant associations, it remains unclear whether the deficiencies contribute directly to these outcomes or are a consequence of the underlying disorders. Longitudinal studies are needed to clarify these relationships.

In addition, residual confounding factors, such as diet, physical activity, and socioeconomic status may have influenced the results. Although adjustments were made for key variables, the lack of a detailed dietary assessment limits our ability to isolate the specific effects of vitamin deficiencies from broader lifestyle factors. Lastly, the study did not include a control group of people without psychiatric disorders, which limits the generalizability of the findings to the wider population.

### 4.6. Future Research and Clinical Implications

The findings from our study highlight the need for routine screening of vitamin D, folate, and B12 levels in psychiatric populations. Given the high prevalence of these deficiencies and their profound impact on both mental and physical health, early detection and correction should be a priority in psychiatric care. This approach has the potential to prevent the worsening of psychiatric symptoms, reduce the risk of long-term metabolic complications, and improve the overall effectiveness of psychiatric treatments.

Future research should focus on large-scale, longitudinal studies that investigate the long-term effects of vitamin supplementation on psychiatric and metabolic outcomes. These studies should explore the interactions between multiple micronutrients, as deficiencies often coexist and interact to exacerbate psychiatric symptoms. A more comprehensive approach that addresses several micronutrient deficiencies simultaneously may offer greater benefits than targeting individual nutrients in isolation.

In addition, the potential for personalized nutrition in psychiatry represents an exciting new frontier. By tailoring nutritional interventions based on individual genetic, metabolic, and environmental factors, clinicians could optimize treatment outcomes and improve both mental and physical health. This approach would allow for more precise interventions that address the unique needs of each patient, offering a more holistic approach to the management of psychiatric disorders.

## 5. Conclusions

Our study highlights the significant role that vitamin D, B9, and B12 deficiencies play in the mental and metabolic health of patients with schizophrenia, major depressive disorder, and bipolar disorder. These deficiencies not only exacerbate psychiatric symptoms, but also contribute to the physical comorbidities that are prevalent in these populations. By addressing these deficiencies through targeted nutritional interventions, clinicians can improve both mental and physical health outcomes and provide a more comprehensive approach to the treatment of psychiatric disorders. The integration of nutritional psychiatry into routine care practices has the potential to revolutionize the management of severe mental illness, offering patients a more holistic and effective treatment strategy.

## Figures and Tables

**Table 1 nutrients-17-01167-t001:** Hypovitaminosis D in schizophrenia.

Schizophrenia	All	Univariate Analysis			Multivariate Analysis	
		Hypovitaminosis D			OR ^a^ (95% CI ^b^) or Standardized Betas	*p*-Value Adjusted
		No	Yes	*p*-Value		
	n = 463	n = 371 (80.1%)	n = 92 (19.9%)			
Sociodemograhics						
Sex						
Women	124 (26.8%)	99 (26.7%)	25 (27.2%)			
Men	339 (73.2%)	272 (73.3%)	67 (72.8%)	1.000		
Age (years)	34.30 (11.61)	33.58 (11.23)	35.24 (10.28)	0.197		
65 years old and older	3 (0.6%)	3 (0.8%)	0 (0%)	1.000		
Labor force status	54 (11.9%)	51 (14.0%)	3 (3.3%)	0.002	0.205 (0.062–0.673)	0.009
Single	392 (86.2%)	316 (86.8%)	76 (83.5%)	0.401		
Education level	166 (65.1%)	137 (64.6%)	29 (67.4%)	0.861		
Psychiatric comorbidities						
ADHD ^c^	3 (0.7%)	3 (0.7%)	0 (0%)	1.000		
Agoraphobia	60 (13.0%)	36 (9.7%)	24 (26.7%)	<0.001	3.417 (1.908–6.119)	<0.001
Generalized anxiety disorder	79 (17.2%)	59 (15.9%)	20 (22.2%)	0.163	1.506 (0.851–2.666)	0.160
Panic disorder	63 (13.7%)	51 (13.8%)	12 (13.3%)	1.000		
Social phobia	54 (11.7%)	40 (74.1%)	14 (15.6%)	0.206		
PTSD ^d^	7 (1.5%)	7 (1.9%)	0 (0%)	0.215		
Addictive comorbidities						
Tobacco smoking	260 (56.4%)	198 (53.7%)	62 (67.4%)	0.019	1.856 (1.139–3.025)	0.013
Tobacco (packs per year)	11.07 (14.20)	10.37 (13.83)	12.93 (14.53)	0.167	0.041 (−1.978–4.691)	0.424
Cannabis consumption	85 (18.4%)	66 (17.8%)	19 (20.7%)	0.549		
Alcohol use disorder	59 (12.9%)	44 (12.0%)	15 (16.3%)	0.296		
Clinical characteristics						
CDSS ^e^ score	4.27 (4.79)	3.84 (4.48)	5.88 (5.48)	0.003	0.175 (0.879–3.233)	0.001
Depression (CDSS ^e^ cutoff)	110 (29.3%)	79 (26.5%)	31 (40.3%)	0.024	1.894 (1.118–3.208)	0.018
SQoL-18 ^f^ Index	54.10 (18.33)	54.16 (18.59)	50.25 (17.95)	0.084	−0.082 (−8.136–0.751)	0.103
Fagerström score	5.08 (2.49)	4.84 (2.60)	5.70 (2.26)	0.055	0.104 (−0.210–1.532)	0.136
GAF ^g^ score	51.47 (15.70)	52.61 (15.04)	45.51 (14.76)	<0.001	−0.184 (−10.531–−3.354)	<0.001
Functionally remitted (GAF ^g^)	97 (23.4%)	83 (25.3%)	14 (16.3%)	0.087	0.582 (0.311–1.088)	0.090
STAI-YA ^h^ score	46.00 (14.49)	50.08 (11.40)	42.33 (15.50)	0.334		
MARS ^i^ score	6.45 (2.27)	6.37 (2.19)	6.54 (2.24)	0.507		
SF-36 ^j^ physical health score	48.37 (9.59)	47.94 (9.03)	45.86 (13.82)	0.582		
SF-36 ^j^ mental health score	32.54 (11.92)	32.43 (10.39)	34.35 (12.49)	0.660		
SBQ-R ^k^ score	7.67 (4.64)	7.59 (4.60)	8.60 (5.04)	0.111	0.088 (−0.257–2.222)	0.120
SBQ-R ^k^ cutoff	143 (46.1%)	103 (43.5%)	40 (54.8%)	0.107	1.570 (0.921–2.674)	0.097
CGI ^l^ score	4.23 (1.15)	4.20 (1.11)	4.48 (1.08)	0.040	0.098 (0.009–0.540)	0.043
UKU I ^m^	4.82 (3.87)	4.77 (3.79)	5.59 (4.25)	0.138	0.091 (−0.235–1.922)	0.125
UKU II ^m^	1.22 (1.50)	1.13 (1.42)	1.47 (1.46)	0.091	0.103 (−0.049–0.743)	0.085
UKU III ^m^	2.46 (2.50)	2.46 (2.46)	2.58 (2.30)	0.726		
UKU IV ^m^	3.26 (3.62)	3.35 (3.88)	3.30 (3.07)	0.935		
Treatments						
Chlorpromazine-equivalent dose	790.84 (756.36)	789.96 (719.19)	814.05 (826.50)	0.780		
Atypical antipsychotics	401 (86.6%)	324 (87.3%)	77 (83.7%)	0.392		
Typical antipsychotics	73 (15.8%)	59 (15.9%)	14 (15.2%)	1.000		
Antipsychotics (typical and atypical)	417 (90.1%)	335 (90.3%)	82 (89.1%)	0.700		
Antidepressants	128 (27.6%)	94 (25.3%)	34 (37.0%)	0.037	1.700 (1.046–2.762)	0.032
Benzodiazepines	134 (28.9%)	106 (28.6%)	28 (30.4%)	0.797		
Mood stabilizers	42 (9.1%)	34 (9.2%)	8 (8.7%)	1.000		
Physical health						
Body mass index	25.87 (5.22)	25.62 (5.04)	26.90 (5.94)	0.060	0.086 (−0.055–2.307)	0.062
Obesity	99 (21.4%)	71 (19.1%)	28 (30.4%)	0.023	1.789 (1.064–3.007)	0.028
Total cholesterol	5.40 (8.59)	5.54 (10.50)	5.40 (1.29)	0.902		
LDL ^n^ cholesterol	3.11 (1.03)	3.05 (0.98)	3.50 (1.22)	<0.001	0.156 (0.161–0.648)	0.001
HDL ^o^ cholesterol	1.40 (0.60)	1.44 (0.60)	1.33 (0.65)	0.149	−0.080 (−0.259–0.013)	0.077
hsCRP ^p^	2.35 (2.31)	2.16 (2.18)	2.67 (2.53)	0.078	0.079 (−0.098–1.015)	0.106
Elevated hsCRP ^p^	258 (62.6%)	208 (61.9%)	50 (65.8%)	0.600		
TSH ^q^	2.32 (1.31)	2.30 (1.37)	2.50 (1.26)	0.212		
Prolactin	618.89 (877.87)	660.52 (963.73)	539.67 (612.89)	0.280		
Vitamin B12	353.78 (128.00)	356.96 (125.27)	348.46 (146.30)	0.697		
Vitamin B9	14.41 (6.87)	14.42 (7.15)	12.99 (5.45)	0.227		
High blood pressure, diagnosed	17 (3.7%)	13 (3.5%)	4 (4.3%)	0.452		
Diabetes	14 (3.0%)	10 (2.7%)	4 (4.3%)	0.300		
High blood pressure, measured	177 (38.4%)	132 (35.8%)	45 (48.9%)	0.023	1.688 (1.057–2.697)	0.028
Hyperglycemia	54 (11.7%)	39 (10.5%)	15 (16.3%)	0.146	1.587 (0.824–3.054)	0.167
Hypertriglyceridemia	124 (27.0%)	88 (23.9%)	36 (39.1%)	0.006	2.171 (1.315–3.586)	0.002
Low HDL ^o^ cholesterol	122 (27.1%)	88 (24.4%)	34 (37.8%)	0.017	2.003 (1.217–3.295)	0.006
High abdominal perimeter	282 (61.7%)	223 (60.9%)	59 (64.8%)	0.548		
Metabolic syndrome	120 (26.1%)	86 (23.3%)	34 (37.8%)	0.007	1.969 (1.200–3.231)	0.007

Note: ^a^ odds ratios; ^b^ confidence interval; ^c^ attention deficit and hyperactivity disorder; ^d^ post-traumatic stress disorder; ^e^ Calgary Depression Scale for Schizophrenia; ^f^ Schizophrenia Quality of Life—18 items; ^g^ Global Assessment of Functioning; ^h^ State-Trait Anxiety Inventory—YA form; ^i^ Medication Adherence Rating Scale; ^j^ 36-Item Short Form Health Survey Questionnaire; ^k^ Suicide Behaviors Questionnaire—Revised; ^l^ clinical global impression; ^m^ Udvalg for Kliniske Undersøgelser; ^n^ low-density lipoprotein; ^o^ high-density lipoprotein; ^p^ high-sensitivity C-reactive protein; ^q^ thyroid-stimulating hormone. Significant values are in blue.

**Table 2 nutrients-17-01167-t002:** Hypovitaminosis D in major depressive disorder.

Major Depressive Disorder	All	Univariate Analysis			Multivariate Analysis	
		Hypovitaminosis D			OR ^a^ (95% CI ^b^) or Standardized Betas	*p*-Value Adjusted
		No	Yes	*p*-Value		
	n = 427	n = 377 (88.3%)	n = 50 (11.7%)			
Sociodemograhics						
Sex						
Women	237 (55.5%)	211 (56.0%)	26 (52.0%)			
Men	190 (44.5%)	166 (44.0%)	24 (48.0%)	0.651		
Age	43.50 (15.06)	43.57 (15.00)	45.97 (15.80)	0.314		
65 years old and older	36 (8.4%)	31 (8.2%)	5 (10.0%)	0.595		
Labor force status	116 (27.6%)	111 (29.9%)	5 (10.2%)	0.003	0.270 (0.104–0.701)	0.007
Single	193 (45.8%)	172 (46.2%)	21 (42.9%)	0.761		
Education level	239 (64.8%)	209 (64.7%)	30 (65.2%)	1.000		
Psychiatric comorbidities						
ADHD ^c^	59 (13.8%)	57 (15.1%)	2 (4.0%)	0.018	0.232 (0.053–1.011)	0.052
Agoraphobia	80 (18.8%)	62 (16.5%)	18 (36.0%)	0.002	2.930 (1.540–5.573)	0.001
Generalized anxiety disorder	215 (50.7%)	194 (51.7%)	21 (42.9%)	0.288		
Panic disorder	117 (27.5%)	99 (26.4%)	18 (36.0%)	0.177	1.588 (0.850–2.968)	0.147
Social phobia	83 (19.5%)	69 (18.4%)	14 (28.0%)	0.128	1.807 (0.914–3.572)	0.089
PTSD ^d^	45 (10.6%)	41 (10.9%)	4 (8.0%)	0.366		
Addictive comorbidities						
Tobacco smoking	181 (47.5%)	163 (47.9%)	18 (43.9%)	0.741		
Tobacco (packs per year)	18.59 (15.77)	17.80 (15.73)	15.35 (13.80)	0.508		
Cannabis consumption	65 (16.8%)	61 (17.7%)	4 (9.8%)	0.143	0.477 (0.158–1.437)	0.188
Alcohol use disorder	64 (16.8%)	59 (17.4%)	5 (12.2%)	0.511		
Clinical characteristics						
CDSS ^e^ score	9.61 (5.56)	9.78 (5.58)	9.14 (5.57)	0.580		
Depression (CDSS ^e^ cutoff)	105 (73.4%)	85 (74.6%)	20 (60.9%)	0.638		
SQoL-18 ^f^ index	42.52 (18.61)	42.61 (19.02)	43.48 (20.61)	0.819		
Fagerström score	4.36 (3.23)	4.31 (3.10)	4.75 (3.60)	0.558		
GAF ^g^ score	55.06 (15.33)	54.76 (14.65)	55.49 (18.17)	0.807		
Functionally remitted (GAF ^g^)	77 (30.8%)	62 (30.0%)	15 (34.9%)	0.587		
STAI-YA ^h^ score	50.30 (13.04)	50.28 (12.90)	49.81 (14.85)	0.830		
MARS ^i^ score	5.81 (2.32)	5.75 (2.30)	6.57 (2.41)	0.028	0.117 (0.044–1.482)	0.038
SF-36 ^j^ physical health score	47.12 (13.37)	48.53 (13.17)	42.50 (14.11)	0.014	−0.150 (−10.781–−1.321)	0.012
SF-36 ^j^ mental health score	27.88 (13.54)	27.75 (13.47)	31.54 (15.71)	0.130	0.088 (1.307–8.546)	0.149
SBQ-R ^k^ score	9.73 (5.31)	9.79 (5.25)	9.13 (5.57)	0.468		
SBQ-R ^k^ cutoff	141 (60.3%)	119 (61.3%)	22 (55.0%)	0.481		
CGI ^l^ score	4.21 (1.25)	4.24 (1.24)	4.14 (1.32)	0.653		
UKU I ^m^	6.74 (6.04)	6.82 (6.43)	5.94 (4.72)	0.455		
UKU II ^m^	1.17 (1.71)	0.97 (1.59)	1.85 (1.91)	0.018	0.201 (0.252–1.510)	0.006
UKU III ^m^	3.29 (3.06)	3.22 (2.90)	3.21 (2.58)	0.995		
UKU IV ^m^	4.26 (4.71)	4.16 (4.99)	4.12 (3.58)	0.963		
Treatments						
Chlorpromazine-equivalent dose	80.04 (228.83)	80.42 (227.27)	92.10 (284.12)	0.741		
Atypical antipsychotics	68 (15.9%)	59 (15.6%)	9 (18.0%)	0.681		
Typical antipsychotics	16 (3.7%)	14 (3.7%)	2 (4.0%)	1.000		
Antipsychotics (typical and atypical)	79 (18.5%)	69 (18.3%)	10 (20.0%)	0.846		
Antidepressants	278 (65.1%)	244 (64.7%)	34 (68.0%)	0.753		
Benzodiazepines	135 (31.6%)	119 (31.6%)	16 (32.0%)	1.000		
Mood stabilizers	15 (3.5%)	15 (4.0%)	0 (0%)	0.149		
Physical health						
Body mass index	25.45 (5.99)	25.23 (5.57)	26.81 (7.02)	0.133	0.083 (−0.203–3.191)	0.084
Obesity	76 (17.9%)	64 (17.1%)	12 (24.0%)	0.240		
Total cholesterol	5.20 (1.16)	5.20 (1.17)	5.47 (1.22)	0.130	0.059 (−0.109–0.542)	0.191
LDL ^n^ cholesterol	3.16 (1.06)	3.15 (1.00)	3.31 (1.39)	0.458		
HDL ^o^ cholesterol	1.53 (0.47)	1.55 (0.46)	1.61 (0.55)	0.458		
hsCRP ^p^	2.11 (2.26)	1.90 (2.07)	2.94 (3.07)	0.034	0.148 (0.356–1.741)	0.003
Elevated hsCRP ^p^	213 (53.7%)	188 (53.3%)	25 (56.8%)	0.749		
TSH ^q^	2.15 (1.66)	2.05 (1.16)	2.26 (1.60)	0.244		
Prolactin	340.79 (365.05)	333.50 (368.12)	379.24 (386.20)	0.438		
Vitamin B12	336.63 (138.91)	337.29 (138.99)	323.54 (130.47)	0.641		
Vitamin B9	16.22 (9.27)	16.51 (9.38)	11.66 (3.72)	<0.001	−0.146 (−8.398–−1.131)	0.010
High blood pressure, diagnosed	48 (11.3%)	39 (10.4%)	9 (18.0%)	0.148	1.725 (0.732–4.064)	0.213
Diabetes	19 (4.5%)	14 (3.7%)	5 (10.0%)	0.059	2.679 (0.909–7.893)	0.074
High blood pressure, measured	156 (36.7%)	132 (35.2%)	24 (48.0%)	0.087	1.589 (0.850–2.972)	0.147
Hyperglycemia	49 (11.6%)	44 (11.8%)	5 (10.2%)	1.000		
Hypertriglyceridemia	75 (17.9%)	60 (16.2%)	15 (30.6%)	0.018	2.251 (1.135–4.463)	0.020
Low HDL ^o^ cholesterol	65 (15.6%)	56 (15.2%)	9 (18.4%)	0.534		
High abdominal perimeter	261 (63.5%)	227 (62.5%)	34 (70.8%)	0.338		
Metabolic syndrome	75 (18.0%)	62 (16.8%)	13 (27.1%)	0.108	1.761 (0.859–3.610)	0.122

Note: ^a^ odds ratios; ^b^ confidence interval; ^c^ attention deficit and hyperactivity disorder; ^d^ post-traumatic stress disorder; ^e^ Calgary Depression Scale for Schizophrenia; ^f^ Schizophrenia Quality of Life—18 items; ^g^ Global Assessment of Functioning; ^h^ State-Trait Anxiety Inventory—YA form; **^i^** Medication Adherence Rating Scale; ^j^ 36-Item Short Form Health Survey Questionnaire; ^k^ Suicide Behaviors Questionnaire—Revised; ^l^ clinical global impression; ^m^ Udvalg for Kliniske Undersøgelser; ^n^ low-density lipoprotein; ^o^ high-density lipoprotein; ^p^ high-sensitivity C-reactive protein; ^q^ thyroid-stimulating hormone. Significant values are in blue.

**Table 3 nutrients-17-01167-t003:** Hypovitaminosis D in bipolar disorder.

Bipolar Disorder	All	Univariate Analysis			Multivariate Analysis	
		Hypovitaminosis D			OR ^a^ (95% CI ^b^) or Standardized Betas	*p*-Value Adjusted
		No	Yes	*p*-Value		
	n = 113	n = 103 (91.2%)	n = 10 (8.8%)			
Sociodemographics						
Sex						
Women	65 (57.5%)	59 (57.3%)	6 (60.0%)			
Men	48 (42.5%)	44 (42.7%)	4 (40.0%)	0.572		
Age	45.37 (14.10)	44.92 (14.13)	55.03 (11.55)	0.031	0.202 (0.890–19.196)	0.032
65 years old and older	10 (8.8%)	8 (7.8%)	2 (20.0%)	0.216		
Labor force status	26 (23.0%)	25 (24.3%)	1 (10.0%)	0.279		
Single	49 (43.4%)	46 (44.7%)	3 (30.0)	0.292		
Education level	65 (66.3%)	57 (64.0%)	8 (88.9%)	0.126	7.934 (0.873–72.135)	0.066
Psychiatric comorbidities						
ADHD ^c^	10 (8.8%)	10 (9.7%)	0 (0%)	0.380		
Agoraphobia	28 (25.7%)	24 (24.0%)	4 (44.4%)	0.170	3.220 (0.733–14.143)	0.121
Generalized anxiety disorder	49 (45.0%)	44 (44.0%)	5 (55.6%)	0.373		
Panic disorder	28 (25.7%)	26 (26.0%)	2 (22.2%)	0.581		
Social phobia	33 (30.3%)	31 (31.0%	2 (22.2%)	0.450		
PTSD ^d^	9 (8.3%)	9 (9.0%)	0 (0%)	0.446		
Addictive comorbidities						
Tobacco smoking	58 (57.4%)	54 (59.3%)	4 (40.0%)	0.201		
Tobacco (packs per year)	20.07 (14.94)	16.88 (12.91)	38.00 (18.42)	0.004	0.547 (0.135–2.212)	0.398
Cannabis consumption	21 (21.2%)	20 (22.5%)	1 (10.0%)	0.327		
Alcohol use disorder	19 (19.6%)	19 (21.8%)	0 (0%)	0.100	<0.001 (–)	-
Clinical characteristics						
CDSS ^e^ score	7.21 (5.95)	7.66 (6.03)	10.00 (0.00)	0.017	0.161 (−4.546–13.465)	0.323
Depression (CDSS ^e^ cutoff)	25 (58.1%)	23 (56.1%)	2 (100%)	0.332		
SQoL-18 ^f^ index	43.62 (21.68)	43.79 (22.65)	28.72 (6.55)	0.002	−0.190 (−36.820–4.498)	0.123
Fagerström score	4.81 (3.23)	4.77 (3.28)	7.00 (2.71)	0.190	0.215 (−0.665–6.268)	0.111
GAF ^g^ score	58.37 (14.43)	58.71 (14.45)	52.29 (13.16)	0.261		
Functionally remitted (GAF ^g^)	35 (43.8%)	33 (45.2%)	2 (28.6%)	0.333		
STAI-YA ^h^ score	47.92 (16.09)	49.65 (15.27)	46.63 (15.90)	0.595		
MARS ^i^ score	6.10 (2.32)	5.93 (2.59)	6.50 (2.80)	0.468		
SF-36 ^j^ physical health score	45.91 (13.61)	46.35 (15.02)	41.26 (8.01)	0.349		
SF-36 ^j^ mental health score	29.73 (15.30)	29.45 (15.63)	26.71 (14.99)	0.636		
SBQ-R ^k^ score	10.01 (5.40)	10.15 (5.24)	9.57 (7.12)	0.787		
SBQ-R ^k^ cutoff	53 (66.2%)	49 (67.1%)	4 (57.1%)	0.439		
CGI ^l^ score	3.96 (1.19)	3.92 (1.19)	4.38 (1.06)	0.300		
UKU I ^m^	6.21 (4.98)	6.40 (5.15)	5.80 (4.92)	0.803		
UKU II ^m^	1.00 (1.27)	1.04 (1.34)	0.80 (0.84)	0.699		
UKU III ^m^	3.30 (3.10)	3.48 (3.28)	3.40 (2.19)	0.957		
UKU IV ^m^	4.95 (4.66)	4.98 (4.77)	6.80 (5.81)	0.427		
Treatments						
Chlorpromazine-equivalent dose	175.16 (345.90)	170.15 (365.35)	282.50 (324.05)	0.351		
Atypical antipsychotics	35 (31.0%)	29 (28.2%)	6 (60.0%)	0.047	4.512 (1.114–18.275)	0.035
Typical antipsychotics	3 (2.7%)	2 (1.9%)	1 (10.0%)	0.245		
Antipsychotics (typical and atypical)	36 (31.9%)	30 (29.1%)	6 (60.0%)	0.054	4.391 (1.088–17.731)	0.038
Antidepressants	71 (62.8%)	64 (62.1%)	7 (70.0%)	0.451		
Benzodiazepines	39 (34.5%)	33 (32.0%)	6 (60.0%)	0.079	2.745 (0.695–10.852)	0.150
Mood stabilizers	42 (37.2%)	38 (36.9%)	4 (40.0%)	0.549		
Physical health						
Body mass index	25.52 (5.45)	25.42 (5.09)	27.33 (6.15)	0.291		
Obesity	22 (20.2%)	18 (18.0%)	4 (44.4%)	0.079	2.972 (0.700–12.610)	0.140
Total cholesterol	5.33 (1.10)	5.31 (1.07)	5.72 (1.62)	0.270		
LDL ^n^ cholesterol	3.18 (1.02)	3.17 (0.92)	3.56 (2.13)	0.672		
HDL ^o^ cholesterol	1.53 (0.50)	1.54 (0.50)	1.50 (0.55)	0.802		
hsCRP ^p^	1.92 (1.95)	1.87 (1.89)	2.62 (2.10)	0.242		
Elevated hsCRP ^p^	56 (54.9%)	48 (52.2%)	8 (80.0%)	0.087	2.806 (0.546–14.415)	0.217
TSH ^q^	2.30 (1.41)	2.21 (1.21)	2.20 (1.46)	0.984		
Prolactin	269.61 (209.20)	254.46 (180.25)	240.11 (146.41)	0.818		
Vitamin B12	353.84 (164.66)	364.40 (172.31)	322.67 (160.15)	0.570		
Vitamin B9	17.28 (8.71)	17.85 (9.09)	12.17 (5.27)	0.137	−0.191 (−13.811–1.265)	−0.191
High blood pressure, diagnosed	8 (7.1%)	8 (7.8%)	0 (0%)	0.461		
Diabetes	2 (1.8%)	2 (2.0%)	0 (0%)	0.829		
High blood pressure, measured	40 (35.4%)	35 (34.0%)	5 (50.0%)	0.321		
Hyperglycemia	9 (8.0%)	8 (7.8%)	1 (10.0%)	0.580		
Hypertriglyceridemia	26 (23.2%)	24 (23.5%)	2 (20.0%)	0.579		
Low HDL ^o^ cholesterol	23 (20.7%)	20 (19.8%)	3 (30.0%)	0.342		
High abdominal perimeter	67 (61.5%)	60 (60.6%)	7 (70.0%)	0.414		
Metabolic syndrome	16 (14.2%)	13 (12.6%)	3 (30.0%)	0.150	2.298 (0.504–10.478)	0.282

Note: ^a^ odds ratios; ^b^ confidence interval; ^c^ attention deficit and hyperactivity disorder; ^d^ post-traumatic stress disorder; ^e^ Calgary Depression Scale for Schizophrenia; ^f^ Schizophrenia Quality of Life—18 items; ^g^ Global Assessment of Functioning; ^h^ State-Trait Anxiety Inventory—YA form; ^i^ Medication Adherence Rating Scale; ^j^ 36-Item Short Form Health Survey Questionnaire; ^k^ Suicide Behaviors Questionnaire—Revised; ^l^ clinical global impression; ^m^ Udvalg for Kliniske Undersøgelser; ^n^ low-density lipoprotein; ^o^ high-density lipoprotein; ^p^ high-sensitivity C-reactive protein; ^q^ thyroid-stimulating hormone. Significant values are in blue.

**Table 4 nutrients-17-01167-t004:** Hypovitaminosis B9 in schizophrenia.

Schizophrenia	All	Univariate Analysis			Multivariate Analysis	
		Hypovitaminosis B9			OR ^a^ (95% CI ^b^) or Standardized Betas	*p*-Value Adjusted
		No	Yes	*p*-Value		
	n = 302	n = 220 (72.8%)	n = 82 (27.2%)			
Sociodemographics						
Sex						
Women	80 (26.5%)	66 (30.0%)	14 (17.1%)			
Men	222 (73.5%)	154 (70.0%)	68 (82.9%)	0.015	0.498 (0.261–0.951)	0.035
Age	34.30 (11.61)	10.82 (0.73)	30.73 (10.03)	0.158	−0.068 (−4.348–1.086)	0.238
65 years old and older	2 (0.7%)	2 (0.9%)	0 (0%)	0.530		
Labor force status	38 (12.6%)	29 (13.2%)	9 (11.1%)	0.700		
Single	268 (88.7%)	192 (87.3%)	76 (92.7%)	0.223		
Education level	129 (61.1%)	102 (67.5%)	27 (45.0%)	0.003	0.401 (0.213–0.754)	0.005
Psychiatric comorbidities						
ADHD ^c^	4 (1.3%)	2 (0.9%)	2 (2.5%)	0.295		
Agoraphobia	40 (13.3%)	33 (15.1%)	7 (8.5%)	0.181	0.514 (0.216–1.221)	0.132
Generalized anxiety disorder	55 (18.3%)	45 (20.6%)	10 (12.2%)	0.097	0.564 (0.268–1.190)	0.133
Panic disorder	43 (14.3%)	37 (17.0%)	6 (7.3%)	0.041	0.386 (0.155–0.958)	0.040
Social phobia	44 (14.7%)	38 (17.4%)	6 (7.3%)	0.028	0.364 (0.147–0.904)	0.029
PTSD ^d^	7 (2.3%)	4 (1.8%)	3 (3.7%)	0.292		
Addictive comorbidities						
Tobacco smoking	163 (54.2%)	114 (52.1%)	49 (59.8%)	0.245		
Tobacco (packs per year)	11.07 (14.20)	8.74 (12.93)	9.94 (9.89)	0.554		
Cannabis consumption	60 (19.9%)	43 (19.5%)	17 (20.7%)	0.871		
Alcohol use disorder	31 (10.4%)	22 (10.1%)	9 (11.2%)	0.831		
Clinical characteristics						
CDSS ^e^ score	4.27 (4.79)	4.45 (5.18)	3.71 (4.31)	0.328		
Depression (CDSS ^e^ cutoff)	67 (29.6%)	53 (31.7%)	14 (23.7%)	0.320		
SQoL-18 ^f^ index	54.10 (18.33)	52.58 (18.68)	53.68 (18.27)	0.683		
Fagerström score	5.08 (2.49)	4.86 (2.38)	4.91 (2.44)	0.907		
GAF ^g^ score	51.47 (15.70)	51.64 (15.67)	53.15 (16.09)	0.499		
Functionally remitted (GAF ^g^)	69 (26.0%)	49 (24.9%)	20 (29.4%)	0.522		
STAI-YA ^h^ score	46.00 (14.49)	48.89 (12.32)	43.00 (10.44)	0.478		
MARS ^i^ score	6.45 (2.27)	6.22 (2.33)	6.47 (2.22)	0.459		
SF-36 ^j^ physical health score	48.37 (9.59)	49.55 (8.11)	49.47 (8.15)	0.977		
SF-36 ^j^ mental health score	32.54 (11.92)	33.49 (13.04)	32.31 (10.86)	0.802		
SBQ-R ^k^ score	7.67 (4.64)	8.10 (4.64)	7.10 (5.06)	0.236		
SBQ-R ^k^ cutoff	90 (47.6%)	74 (49.7%)	16 (40.0%)	0.291		
CGI ^l^ score	4.23 (1.15)	4.20 (1.14)	4.31 (1.15)	0.510		
UKU I ^m^	4.82 (3.87)	5.95 (3.87)	4.60 (4.46)	0.060	−0.149 (−2.799–0.009)	0.052
UKU II ^m^	1.22 (1.50)	1.24 (1.52)	1.24 (1.65)	1.000		
UKU III ^m^	2.46 (2.50)	3.03 (2.56)	2.57 (2.69)	0.321		
UKU IV ^m^	3.26 (3.62)	4.68 (4.06)	3.52 (3.84)	0.106	−0.122 (−2.543–0.291)	0.119
Treatments						
Chlorpromazine-equivalent dose	790.84 (756.36)	729.10 (635.60)	840.24 (867.42)	0.292		
Atypical antipsychotics	258 (85.4%)	158 (84.1%)	73 (89.0%)	0.360		
Typical antipsychotics	50 (16.6%)	33 (15.0%)	17 (20.7%)	0.229		
Antipsychotics (typical and atypical)	269 (89.1%)	195 (88.6%)	74 (90.2%)	0.836		
Antidepressants	85 (28.1%)	68 (30.9%)	17 (20.7%)	0.086	0.548 (0.296–1.015)	0.056
Benzodiazepines	90 (29.8%)	66 (30.0%)	24 (29.3%)	1.000		
Mood stabilizers	39 (12.9%)	27 (12.3%)	12 (14.6%)	0.568		
Physical health						
Body mass index	25.87 (5.22)	25.43 (5.05)	25.96 (6.11)	0.444		
Obesity	60 (19.9%)	42 (19.1%)	18 (22.0%)	0.627		
Total cholesterol	5.40 (8.59)	4.97 (1.11)	4.84 (1.22)	0.383		
LDL ^n^ cholesterol	3.11 (1.03)	3.06 (1.00)	2.99 (1.04)	0.619		
HDL ^o^ cholesterol	1.40 (0.60)	3.06 (1.00)	2.99 (1.04)	0.619		
hsCRP ^p^	2.35 (2.31)	1.94 (1.86)	2.38 (2.44)	0.179	0.095 (−0.112–1.003)	0.117
Elevated hsCRP ^p^	165 (59.4%)	121 (58.2%)	44 (62.9%)	0.574		
TSH ^q^	2.32 (1.31)	2.31 (1.34)	2.50 (1.47)	0.294		
Prolactin	618.89 (877.87)	618.15 (947.97)	676.19 (831.35)	0.633		
Vitamin D	52.57 (30.40)	59.55 (32.28)	59.72 (32.04)	0.967		
Vitamin B12	353.78 (128.00)	366.16 (130.69)	324.97 (109.37)	0.008	−0.140 (−72.334–−6.530)	0.019
High blood pressure, diagnosed	9 (3.0%)	7 (3.2%)	2 (2.5%)	0.541		
Diabetes	11 (3.7%)	10 (4.6%)	1 (1.2%)	0.152	0.294 (0.036–2.408)	0.254
High blood pressure, measured	109 (36.3%)	82 (37.6%)	27 (32.9%)	0.502		
Hyperglycemia	33 (10.9%)	25 (11.4%)	8 (9.8%)	0.836		
Hypertriglyceridemia	80 (26.8%)	55 (25.1%)	25 (31.2%)	0.304		
Low HDL ^o^ cholesterol	87 (29.4%)	59 (27.2%)	28 (35.4%)	0.194	1.494 (0.853–2.616)	0.160
High abdominal perimeter	168 (56.4%)	122 (55.7%)	46 (58.2%)	0.791		
Metabolic syndrome	80 (26.6%)	56 (25.5%)	24 (29.6%)	0.466		

Note: ^a^ odds ratios; ^b^ confidence interval; ^c^ attention deficit and hyperactivity disorder; ^d^ post-traumatic stress disorder; ^e^ Calgary Depression Scale for Schizophrenia; ^f^ Schizophrenia Quality of Life—18 items; ^g^ Global Assessment of Functioning; ^h^ State-Trait Anxiety Inventory—YA form; ^i^ Medication Adherence Rating Scale; ^j^ 36-Item Short Form Health Survey Questionnaire; ^k^ Suicide Behaviors Questionnaire—Revised; ^l^ clinical global impression; ^m^ Udvalg for Kliniske Undersøgelser; ^n^ low-density lipoprotein; ^o^ high-density lipoprotein; ^p^ high-sensitivity C-reactive protein; ^q^ thyroid-stimulating hormone. Significant values are in blue.

**Table 5 nutrients-17-01167-t005:** Hypovitaminosis B9 in major depressive disorder.

Major Depressive Disorder	All	Univariate Analysis			Multivariate Analysis	
		Hypovitaminosis B9			OR ^a^ (95% CI ^b^) or Standardized Betas	*p*-Value Adjusted
		No	Yes	*p*-Value		
	n = 315	n = 227 (72.1%)	n = 88 (27.9%)			
Sociodemographics						
Sex						
Women	174 (55.2%)	132 (58.1%)	42 (47.7%)			
Men	141 (44.8%)	95 (41.9%)	46 (52.3%)	0.102	−0.063 (−0.196–0.056)	0.275
Age	43.50 (15.06)	45.49 (16.13)	36.66 (11.80)	<0.001	−0.245 (−12.173–−4.742)	<0.001
65 years old and older	28 (8.9%)	27 (11.9%)	1 (1.1%)	0.001	-	-
Labor force status	78 (25.0%)	51 (22.5%)	27 (31.8%)	0.106	1.545 (0.871–2.741)	0.137
Single	144 (46.2%)	90 (40.0%)	54 (62.1%)	0.001	1.312 (0.711–2.418)	0.385
Education level	174 (65.9%)	126 (65.3%)	48 (67.6%)	0.771		
Psychiatric comorbidities						
ADHD ^c^	44 (14.0%)	23 (10.1%)	21 (23.9%)	0.003	2.082 (1.055–4.108)	0.034
Agoraphobia	60 (19.2%)	42 (18.7%)	18 (20.5%)	0.750		
Generalized anxiety disorder	157 (50.3%)	108 (48.0%)	49 (56.3%)	0.208		
Panic disorder	85 (27.2%)	63 (28.0%)	22 (25.0%)	0.672		
Social phobia	60 (19.2%)	45 (20.0%)	15 (17.0%)	0.633		
PTSD ^d^	33 (10.5%)	23 (10.2%)	10 (11.4%)	0.838		
Addictive comorbidities						
Tobacco smoking	127 (43.1%)	85 (40.3%)	42 (50.0%)	0.152	1.257 (0.741–2.131)	0.397
Tobacco (packs per year)	18.59 (15.77)	18.21 (17.37)	15.15 (14.10)	0.369		
Cannabis consumption	52 (17.3%)	32 (14.9%)	20 (23.5%)	0.090	1.265 (0.655–2.442)	0.484
Alcohol use disorder	41 (13.9%)	28 (13.3%)	13 (15.5%)	0.709		
Clinical characteristics						
CDSS ^e^ score	9.61 (5.56)	9.11 (5.91)	11.09 (4.91)	0.293		
Depression (CDSS ^e^ cutoff)	61 (71.8%)	51 (68.9%)	10 (90.9%)	0.121	2.803 (0.309–25.437)	0.360
SQoL-18 ^f^ index	42.52 (18.61)	43.30 (19.62)	41.41 (19.09)	0.555		
Fagerström score	4.36 (3.23)	4.16 (3.29)	4.40 (3.12)	0.687		
GAF ^g^ score	55.06 (15.33)	57.31 (14.95)	52.03 (13.86)	0.069	−0.099 (−9.806–2.250)	0.218
Functionally remitted (GAF^g^)	56 (32.0%)	49 (34.3%)	7 (21.9%)	0.212		
STAI-YA ^h^ score	50.30 (13.04)	47.81 (12.46)	51.54 (13.07)	0.073	0.117 (−0.714–7.792)	0.102
MARS ^i^ score	5.81 (2.32)	5.90 (2.39)	5.55 (2.39)	0.350		
SF-36 ^j^ physical health score	47.12 (13.37)	47.53 (16.29)	49.86 (9.94)	0.258		
SF-36 ^j^ mental health score	27.88 (13.53)	30.08 (16.78)	27.92 (11.12)	0.428		
SBQ-R ^k^ score	9.73 (5.31)	9.58 (5.14)	10.28 (5.73)	0.521		
SBQ-R ^k^ cutoff	93 (61.2%)	74 (60.2%)	19 (65.5%)	0.675		
CGI ^l^ score	4.21 (1.25)	4.12 (1.32)	4.39 (0.97)	0.270		
UKU I ^m^	6.74 (6.04)	7.81 (7.32)	5.90 (4.90)	0.259		
UKU II ^m^	1.17 (1.71)	1.25 (1.85)	0.90 (1.34)	0.425		
UKU III ^m^	3.29 (3.06)	3.74 (3.43)	2.76 (3.03)	0.230		
UKU IV ^m^	4.26 (4.71)	4.77 (4.50)	5.05 (5.02)	0.806		
Treatments						
Chlorpromazine-equivalent dose	80.04 (228.83)	76.59 (221.48)	69.66 (192.40)	0.797		
Atypical antipsychotics	51 (16.2%)	37 (16.3%)	14 (15.9%)	1.000		
Typical antipsychotics	9 (2.9%)	5 (2.2%)	4 (4.5%)	0.222		
Antipsychotics (typical and atypical)	57 (18.1%)	40 (17.6%)	17 (19.3%)	0.745		
Antidepressants	202 (64.1%)	152 (67.0%)	50 (56.8%)	0.116	0.921 (0.538–1.576)	0.764
Benzodiazepines	91 (28.9%)	66 (29.1%)	25 (28.4%)	1.000		
Mood stabilizers	8 (2.5%)	8 (3.5%)	0 (0%)	0.070	-	-
Physical health						
Body mass index	25.45 (5.99)	25.34 (6.03)	25.72 (6.54)	0.630		
Obesity	62 (19.8%)	42 (18.6%)	20 (23.0%)	0.429		
Total cholesterol	5.20 (1.16)	5.21 (1.10)	4.99 (1.20)	0.134	0.013 (−0.243–0.307)	0.821
LDL ^n^ cholesterol	3.16 (1.06)	3.17 (0.99)	2.94 (1.13)	0.084	−0.015 (−0.292–0.221)	0.785
HDL ^o^ cholesterol	1.53 (0.47)	3.17 (0.99)	2.94 (1.13)	0.084	−0.081 (−0.194–0.026)	0.132
hsCRP ^p^	2.11 (2.26)	1.98 (2.27)	2.08 (2.30)	0.733		
Elevated hsCRP ^p^	150 (51.2%)	106 (49.5%)	44 (55.7%)	0.360		
TSH ^q^	2.15 (1.66)	1.99 (1.13)	2.12 (1.13)	0.342		
Prolactin	340.79 (365.05)	317.38 (297.58)	337.80 (261.21)	0.596		
Vitamin D	64.96 (32.49)	70.92 (30.26)	63.44 (34.99)	0.068	−0.106 (−15.836–0.970)	0.083
Vitamin B12	336.63 (138.91)	350.69 (149.24)	300.28 (104.49)	0.005	−0.150 (−82.782–−10.810)	0.011
High blood pressure, diagnosed	37 (11.8%)	28 (12.3%)	9 (10.3%)	0.700		
Diabetes	15 (4.8%)	11 (4.8%)	4 (4.6%)	0.595		
High blood pressure, measured	123 (39.3%)	93 (41.2%)	30 (34.5%)	0.303		
Hyperglycemia	43 (13.7%)	35 (15.4%)	8 (9.2%)	0.199	0.858 (0.357–2.057)	0.731
Hypertriglyceridemia	61 (19.7%)	41 (18.1%)	20 (23.8%)	0.265		
Low HDL ^o^ cholesterol	52 (16.8%)	31 (13.8%)	21 (25.0%)	0.026	1.914 (1.000–3.664)	0.050
High abdominal perimeter	182 (60.3%)	139 (64.1%)	43 (50.6%)	0.037	0.912 (0.522–1.592)	0.745
Metabolic syndrome	60 (19.4%)	45 (19.9%)	15 (18.1%)	0.871		

Note: ^a^ odds ratios; ^b^ confidence interval; ^c^ attention deficit and hyperactivity disorder; ^d^ post-traumatic stress disorder; ^e^ Calgary Depression Scale for Schizophrenia; ^f^ Schizophrenia Quality of Life—18 items; ^g^ Global Assessment of Functioning; ^h^ State-Trait Anxiety Inventory—YA form; ^i^ Medication Adherence Rating Scale; ^j^ 36-Item Short Form Health Survey Questionnaire; ^k^ Suicide Behaviors Questionnaire—Revised; ^l^ clinical global impression; ^m^ Udvalg for Kliniske Undersøgelser; ^n^ low-density lipoprotein; ^o^ high-density lipoprotein; ^p^ high-sensitivity C-reactive protein; ^q^ thyroid-stimulating hormone. Significant values are in blue.

**Table 6 nutrients-17-01167-t006:** Hypovitaminosis B9 in bipolar disorder.

Bipolar Disorder	All	Univariate Analysis			Multivariate Analysis	
		Hypovitaminosis B9			OR ^a^ (95% CI ^b^) or Standardized Betas	*p*-Value Adjusted
		No	Yes	*p*-Value		
	n = 84	n = 65 (77.4%)	19 (22.6%)			
Sociodemographics						
Sex						
Women	52 (61.9%)	43 (66.2%)	9 (47.4%)			
Men	32 (38.1%)	22 (33.8%)	10 (47.4%)	0.181	0.483 (0.170–1.371)	0.172
Age	45.37 (14.10)	45.64 (14.58)	42.37 (10.82)	0.368		
65 years old and older	5 (6.0%)	5 (7.7%)	0 (0%)	0.268		
Labor force status	18 (21.4%)	13 (20.0%)	5 (26.3%)	0.540		
Single	39 (46.4%)	31 (47.7%)	8 (42.1%)	0.795		
Education level	51 (70.8%)	38 (67.9%)	13 (81.2%)	0.238		
Psychiatric comorbidities						
ADHD ^c^	4 (4.8%)	2 (3.1%)	2 (10.5%)	0.219		
Agoraphobia	20 (24.4%)	15 (23.4%)	5 (27.8%)	0.759		
Generalized anxiety disorder	37 (45.1%)	29 (45.3%)	8 (44.4%)	1.000		
Panic disorder	20 (24.4%)	14 (21.9%)	6 (33.3%)	0.358		
Social phobia	24 (29.3%)	20 (31.2%)	4 (22.2%)	0.334		
PTSD ^d^	6 (7.3%)	6 (9.4%)	0 (0%)	0.214		
Addictive comorbidities						
Tobacco smoking	36 (48.0%)	27 (46.6%)	9 (52.9%)	0.784		
Tobacco (packs per year)	20.07 (14.94)	16.50 (13.25)	10.29 (7.09)	0.126	−0.241 (−17.411–3.853)	0.201
Cannabis consumption	16 (21.9%)	11 (19.3%)	5 (31.2%)	0.321		
Alcohol use disorder	14 (19.4%)	10 (18.2%)	4 (23.5%)	0.431		
Clinical characteristics						
CDSS ^e^ score	7.21 (5.95)	7.33 (6.64)	5.38 (4.78)	0.449		
Depression (CDSS ^e^ cutoff)	15 (46.9%)	12 (50.0%)	3 (37.5%)	0.421		
SQoL-18 ^f^ index	43.62 (21.68)	42.26 (19.71)	48.53 (27.74)	0.330		
Fagerström score	4.81 (3.23)	5.06 (3.60)	3.00 (2.45)	0.115	−0.264 (−4.910–0.457)	0.101
GAF ^g^ score	58.37 (14.43)	59.80 (16.09)	58.67 (12.10)	0.820		
Functionally remitted (GAF ^g^)	27 (46.6%)	19 (41.3%)	8 (66.7%)	0.107	2.857 (0.705–11.580)	0.141
STAI-YA ^h^ score	47.92 (16.09)	49.27 (13.73)	47.86 (13.25)	0.736		
MARS ^i^ score	6.10 (2.32)	6.25 (2.19)	5.82 (2.19)	0.484		
SF-36 ^j^ physical health score	45.91 (13.61)	44.15 (15.22)	51.20 (15.12)	0.120	0.164 (−3.141–15.100)	0.195
SF-36 ^j^ mental health score	29.73 (15.30)	28.32 (16.86)	33.38 (16.80)	0.311		
SBQ-R ^k^ score	10.01 (5.40)	10.09 (5.32)	9.92 (5.73)	0.922		
SBQ-R ^k^ cutoff	39 (68.4%)	31 (68.9%)	8 (66.7%)	0.570		
CGI ^l^ score	3.96 (1.19)	3.87 (1.33)	3.67 (0.89)	0.528		
UKU I ^m^	6.21 (4.98)	6.68 (5.69)	5.67 (2.29)	0.465		
UKU II ^m^	1.00 (1.27)	1.00 (0.87)	1.00 (1.23)	1.000		
UKU III ^m^	3.30 (3.10)	3.40 (3.42)	3.22 (3.27)	0.893		
UKU IV ^m^	4.95 (4.66)	6.60 (5.33)	5.44 (4.45)	0.566		
Treatments						
Chlorpromazine-equivalent dose	175.16 (345.90)	215.77 (412.32)	190.79 (295.48)	0.806		
Atypical antipsychotics	30 (35.7%)	23 (35.4%)	7 (36.8%)	1.000		
Typical antipsychotics	3 (3.6%)	2 (3.1%)	1 (5.3%)	0.542		
Antipsychotics (typical and atypical)	31 (36.9%)	24 (36.9%)	7 (36.8%)	1.000		
Antidepressants	49 (58.3%)	37 (56.9%)	12 (63.2%)	0.792		
Benzodiazepines	26 (31.0%)	21 (32.3%)	5 (26.3%)	0.780		
Mood stabilizers	34 (40.5%)	26 (40.0%)	8 (42.1%)	1.000		
Physical health						
Body mass index	25.52 (5.45)	25.56 (5.33)	26.50 (4.19)	0.492		
Obesity	17 (20.7%)	12 (18.8%)	5 (27.8%)	0.511		
Total cholesterol	5.33 (1.10)	5.34 (1.10)	5.15 (1.22)	0.538		
LDL ^n^ cholesterol	3.15 (1.02)	3.32 (1.08)	2.91 (1.11)	0.283		
HDL ^o^ cholesterol	1.53 (0.50)	3.32 (1.08)	2.91 (1.11)	0.283		
hsCRP ^p^	1.92 (1.95)	1.80 (1.81)	2.17 (2.10)	0.479		
Elevated hsCRP ^p^	41 (53.2%)	31 (50.8%)	10 (62.5%)	0.575		
TSH ^q^	2.30 (1.41)	2.43 (1.48)	2.71 (1.44)	0.472		
Prolactin	269.61 (209.20)	269.90 (196.22)	268.82 (171.67)	0.984		
Vitamin D	59.21 (27.56)	64.33 (26.74)	53.88 (28.61)	0.167	−0.191 (−27.796–3.037)	0.114
Vitamin B12	353.84 (164.66)	365.35 (179.24)	316.26 (104.37)	0.260		
High blood pressure, diagnosed	6 (7.2%)	6 (9.4%)	0 (0%)	0.199	-	-
Diabetes	1 (1.2%)	1 (1.6%)	0 (0%)	0.771		
High blood pressure, measured	29 (34.5%)	21 (32.3%)	8 (42.1)	0.428		
Hyperglycemia	6 (7.2%)	4 (6.2%)	2 (11.1%)	0.387		
Hypertriglyceridemia	22 (26.8%)	14 (21.9%)	8 (44.4%)	0.073	2.390 (0.731–7.819)	0.150
Low HDL ^o^ cholesterol	20 (24.7%)	13 (20.6%)	7 (38.9%)	0.130	2.430 (0.763–7.744)	0.133
High abdominal perimeter	54 (66.7%)	41 (66.1%)	13 (68.4%)	1.000		
Metabolic syndrome	14 (16.9%)	9 (13.8%)	5 (27.8%)	0.149	3.042 (0.788–11.747)	0.107

Note: ^a^ odds ratios; ^b^ confidence interval; ^c^ attention deficit and hyperactivity disorder; ^d^ post-traumatic stress disorder; ^e^ Calgary Depression Scale for Schizophrenia; ^f^ Schizophrenia Quality of Life—18 items; ^g^ Global Assessment of Functioning; ^h^ State-Trait Anxiety Inventory—YA form; ^i^ Medication Adherence Rating Scale; ^j^ 36-Item Short Form Health Survey Questionnaire; ^k^ Suicide Behaviors Questionnaire—Revised; ^l^ clinical global impression; ^m^ Udvalg for Kliniske Undersøgelser; ^n^ low-density lipoprotein; ^o^ high-density lipoprotein; ^p^ high-sensitivity C-reactive protein; ^q^ thyroid-stimulating hormone.

## Data Availability

The data presented in this study are available on request from the corresponding author.

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
