# Peer review of "Vitamin D, B9, and B12 Deficiencies as Key Drivers of Clinical Severity and Metabolic Comorbidities in Major Psychiatric Disorders"

_nutrients, 2025, doi:10.3390/nu17071167_

Round 1
Reviewer 1 Report
Comments and Suggestions for Authors
The authors present an interesting cross-sectional study in which >1000 participants were who were attending clinical practises for psychiatric disorders were evaluated and profiled for their respective levels of micronutrients vitamin D, B9 and B12. The idea of the study was to correlate levels of these essential micronutrients with incidence and severity of distinct disorders, such as to inform practise of the role of diet/supplementation in addressing such. Overall, the authors found that across the patients populations vitamin deficiency was highly prevalent. Of note, vitamin D deficiency was notably present in those with worse psychiatric states – the same of which could be said for vitamins B9 and B12 albeit to a lesser extent.
In reviewing the manuscript I made a couple of observations. The following should be considered when preparing a suitable revision.
- It would have been beneficial to include a control group to determine whether certain thresholds of these nutrients increase risk/contribute towards the examined disorders. Examining only within populations that had pre-existing conditions is somewhat limited in the scope/conclusions that can be drawn.
- More details on how the levels of each nutrient were measured are required. Were any protocols in place for the sampling i.e. fasting? What kits/assays were used to measure the levels?
- The formatting of the table requires attention. The lack of break lines or spacing makes interpreting some values incredibly difficult. Moreover, some values are ‘bold’ , and it is assumed these are the traits which were initially significant and then scrutinised by multivariate analyses. I think this should be reconsidered to make it stand out more/distinguish it stronger.
- The data obtained is largely clinical, but the discussion is biochemical in its approach. While it is of value to discuss the role these vitamins play at the biochemical level, I was expecting more reference to correlations between levels of each micronutrient and clinical manifestations. While I do appreciate the structure of the discussion, I recommend the authors revisit the discussion pieces balance it by increasing the references to clinical outcomes associated with nutrient levels.
Author Response
Responses to review
We would like to express our sincere gratitude for your thorough and valuable review of our manuscript titled "Vitamin D, B9, and B12 Deficiencies as Key Drivers of Clinical Severity and Metabolic Comorbidities in Major Psychiatric Disorders". Below, we provide a detailed response to each of your recommendations and explain the revisions we have made accordingly (with the exception of the table format change, the proposed revisions are highlighted in yellow in the corrected manuscript).
Reviewer #1
- Unfortunately, we were unable to set up a control group for this study. We thank reviewer #1 and will take this indication into account in future studies, which will bring greater clarity to our methods.
- We have added the available data on nutrient measurement on page 3 of the manuscript, as highlighted in yellow, which will improve its precision, for which we thank the reviewer #1.
- Tables have been reformatted to make them easier to read. Spacing and fonts have been adjusted and we've taken inspiration from other published Nutrients manuscripts to meet your recommendations. Significant values are now blue and bold, this makes it easier to read.
- We would like to thank reviewer #1 for his constructive comments on discussion, which helped us to improve our manuscript. We have kept the discussion format and added a clinical section, to better contextualize and help clinicians.
We greatly appreciate your constructive feedback and believe that the revisions made have significantly strengthened the manuscript. Your suggestions have helped us in improving the quality of our study. All the authors have approved this revision. We hope that the revised version meets your expectations.
Thank you once again for your valuable insights and for helping us to improve our work.
Sincerely,
Theo Korchia and Melanie Faugere
Reviewer 2 Report
Comments and Suggestions for Authors
This manuscript is well-written, meticulously organized, and easily comprehensible, effectively addressing vitamin deficiencies (B9, B12, and D) in major psychiatric disorders. The authors have commendably discussed the epigenetic impacts of vitamins B9 and B12 in the discussion section, providing valuable insights into the consideration of vitamin deficiency in psychiatric patients. The manuscript could be further enhanced by addressing the significant influence of vitamin D on DNA methylation in the discussion section as well. This addition would provide a more comprehensive overview of the epigenetic effects of these vitamins in the context of psychiatric disorders.
Author Response
Responses to review
We would like to express our sincere gratitude for your thorough and valuable review of our manuscript titled "Vitamin D, B9, and B12 Deficiencies as Key Drivers of Clinical Severity and Metabolic Comorbidities in Major Psychiatric Disorders". Below, we provide a detailed response to each of your recommendations and explain the revisions we have made accordingly (with the exception of the table format change, the proposed revisions are highlighted in yellow in the corrected manuscript).
Reviewer #2
We would like to thank reviewer #2 for his encouraging comments. We would like to point out that a number of changes have been made to the manuscript in response to reviewer #1's comments, in particular the formatting of the tables has been improved.
Following your pertinent recommendations, we have also added a paragraph on the significant influence of vitamin D on DNA methylation (references 57, 58, 72-75).
We greatly appreciate your constructive feedback and believe that the revisions made have significantly strengthened the manuscript. Your suggestions have helped us in improving the quality of our study. All the authors have approved this revision. We hope that the revised version meets your expectations.
Thank you once again for your valuable insights and for helping us to improve our work.
Sincerely,
Theo Korchia and Melanie Faugere
Round 2
Reviewer 1 Report
Comments and Suggestions for Authors
The authors have suitably addressed my comments.